# Particle Separation in a Microchannel with a T-Shaped Cross-Section Using Co-Flow of Newtonian and Viscoelastic Fluids

**DOI:** 10.3390/mi14101863

**Published:** 2023-09-28

**Authors:** Jinhyeuk Song, Jaekyeong Jang, Taehoon Kim, Younghak Cho

**Affiliations:** 1Department of Mechanical System Design Engineering, Seoul National University of Science & Technology, 232 Gongneung-ro, Nowon-gu, Seoul 01811, Republic of Korea; toptree2@naver.com; 2Department of Mechanical Design and Robot Engineering, Seoul National University of Science & Technology, 232 Gongneung-ro, Nowon-gu, Seoul 01811, Republic of Korea; jjangy5720@naver.com

**Keywords:** T-shaped cross-section, particle separation, co-flow, Newtonian fluid, viscoelastic fluid

## Abstract

In this study, we investigated the particle separation phenomenon in a microchannel with a T-shaped cross-section, a unique design detailed in our previous study. Utilizing a co-flow system within this T-shaped microchannel, we examined two types of flow configuration: one where a Newtonian fluid served as the inner fluid and a viscoelastic fluid as the outer fluid (Newtonian/viscoelastic), and another where both the inner and outer fluids were Newtonian fluids (Newtonian/Newtonian). We introduced a mixture of three differently sized particles into the microchannel through the outer fluid and observed that the co-flow of Newtonian/viscoelastic fluids effectively separated particles based on their size compared with Newtonian/Newtonian fluids. In this context, we evaluated and compared the particle separation efficiency, recovery rate, and enrichment factor across both co-flow configurations. The Newtonian/viscoelastic co-flow system demonstrated a superior efficiency and recovery ratio when compared with the Newtonian/Newtonian system. Additionally, we assessed the influence of the flow rate ratio between the inner and outer fluids on particle separation within each co-flow system. Our results indicated that increasing the flow rate ratio enhanced the separation efficiency, particularly in the Newtonian/viscoelastic co-flow configuration. Consequently, this study substantiates the potential of utilizing a Newtonian/viscoelastic co-flow system in a T-shaped straight microchannel for the simultaneous separation of three differently sized particles.

## 1. Introduction

Microfluidic devices that facilitate the separation and manipulation of particles or cells through different forces generated during fluid flow within a microchannel have generated growing interest over recent decades [1,2,3]. These devices, known for their ability to manipulate particles without relying on external forces, are characterized by their cost and processing efficiency, as well as minimal sample consumption. Consequently, their applications could span diverse areas, including medicine, chemistry, and biology. The processes of particle focusing and separation are critical for sample preparation in the field of biomedical and chemical analysis [4,5,6]. Various studies have been conducted to focus or separate particles using various forces that act upon particles in the fluid passing through the microchannel.

Inertial and viscoelastic microfluidics have been widely used for practical applications for particle/cell separation and rare-cell enrichment and separation [7,8,9,10,11]. They are simple, label-free, and inexpensive because they can manipulate and separate particles without relying on external force, so various efforts have been made to apply them to the above-mentioned fields. Many research groups have tried to separate and manipulate various biological samples, including circulating tumor cells (CTCs) [10], blood cells [11], and bacteria [1], utilizing either Newtonian or viscoelastic fluid flow. The behaviors of particles due to the inertial force generated by high flow velocity in the microchannel or the elastic force generated by the viscoelasticity of viscoelastic fluid were intensively investigated for various parameters, such as flow rate, particle size, and channel geometry.

Recently, many research groups have used a co-flow of Newtonian and viscoelastic fluids in microchannels to wash or separate particles based on their size [12,13,14,15,16,17,18,19]. D. Yuan et al. [12,13] showed the feasibility of particle separation based on size at a low sample flow rate utilizing the co-flow of Newtonian and viscoelastic fluids in a microfluidic channel with a rectangular cross-section. B. Ha et al. [14] demonstrated kinetic separation and washing of target particles using a co-flow of a non-Newtonian sample fluid and a Newtonian carrier medium. However, they used a non-Newtonian fluid containing λ-DNA, which was expensive and could separate only two kinds of particles. F. Tian et al. reported size-dependent particle separation based on a co-flow system involving Newtonian and viscoelastic fluids that leveraged the interfacial effects between the two fluids [15]. They also presented an interfacial viscoelastic microfluidic system for label-free and size-specific isolation of tumor cells, dependent on the stability of the interface occurring between the sample outer flow (whole blood) and viscoelastic core flow [16]. However, there was a low throughput of the sample treatment due to a low sample-to-sheath flow rate ratio, as well as the low total flow rate of the two fluids. P. Liu et al. [18,19] utilized a viscoelastic microfluidic technique for the length-based separation and enrichment of *B. subtilis* populations, in addition to the shape-based separation and enrichment of *S. Cerevisiae* cells, in a label-free and continuous manner. This work also suffered from the same low throughput of sample treatment as the work of F. Tian [15]. On the other hand, D. Lee et al. [20,21] achieved highly efficient particle separation using the co-flow systems of two miscible liquids with different viscosities. This enabled them to achieve inflection point focusing and alter inertial focusing locations by manipulating complex velocity profiles. Notably, they could control the focusing positions using a co-flow of two Newtonian fluids, which distinguished them from the method used by other research groups mentioned above.

Recently, Raoufi et al. [22] reported that at the obtuse corner of a channel cross-section, the elastic force tends to increase the occurrence of particle focusing. A higher corner angle affects particle focusing, allowing for a more condensed focusing band for smaller particles across a broader range of flow rates. In a previous study [23], a novel yet simple fabrication method was developed for producing a microchannel with a T-shaped cross-section; this method employs soft lithography, which is then followed by a self-alignment procedure involving two PDMS molds. This experimental work demonstrated that in a microfluidic channel with a T-shaped cross-section, particles are effectively focused near the obtuse corner of its cross-section when utilizing a viscoelastic fluid.

In this study, we fabricated a microfluidic device featuring a microchannel with a T-shaped cross-section coupled with a sheath flow microchannel. This device was utilized to conduct experiments on size-dependent particle separation, employing the co-flow of Newtonian/viscoelastic fluids and Newtonian/Newtonian fluids across various flow rate ratios. Furthermore, when compared with existing devices with rectangular microchannels, the proposed microfluidic device demonstrated a noticeable improvement in the separation efficiency, recovery ratio, and enrichment factor.

## 2. Materials and Methods

### 2.1. Design and Fabrication Process of the Microchannel with a T-Shaped Cross-Section

For separation based on particle size, a microfluidic device was fabricated based on the fabrication process of a microfluidic channel with a T-shaped cross-section proposed in a previous study [23]. Figure 1 shows the schematic view of the fabricated microfluidic device, with channel heights (*h*_1_, *h*_2_) of 90 µm and 45 µm, widths (*W*_1_, *W*_2_) of 135 µm and 45 µm, and hydraulic diameter (*D_h_*) of 72 µm, respectively. The width and height of all the outlets are 50 μm and 90 μm, respectively. Outlets 1 and 5 for small particles are located near the channel walls; Outlet 3 for large particles is located near the channel center; and Outlets 2 and 4 for middle particles are located between Outlets 1 and 5 and Outlet 3.

Briefly, the microfluidic device was fabricated through photolithography using an SU-8 2050 photoresist, PDMS (polydimethylsiloxane) molding, self-alignment between PDMS molds, and plasma bonding processes. It consisted of the microchannel with a T-shaped cross-section and the sheath flow microchannel, including two inlets and five outlets.

### 2.2. Sample Preparation and Experimental Procedure

After conducting the particle-focusing experiments with flows of Newtonian and viscoelastic fluids separately, we carried out particle separation experiments across four distinct cases. Table 1 presents the experimental conditions for introducing the sample flow into inlet 1 and the sheath flow into inlet 2 in the microchannel with a T-shaped cross-section, utilizing a combination of co-flows of Newtonian and viscoelastic fluids, as shown in Figure 1a. We conducted particle separation experiments for Case 1 and Case 2, which derived the most effective separation results among the four cases. In Case 1, viscoelastic fluid with sample particles was injected as the sample flow (inlet 1), and Newtonian fluid was injected as the sheath flow (inlet 2). In Case 2, Newtonian fluid with sample particles was injected as the sample flow (inlet 1), and Newtonian fluid was injected as the sheath flow (inlet 2).

Deionized (DI) water served as the Newtonian fluid, while the viscoelastic fluid utilized was a PEO (polyethylene oxide) solution. This PEO solution was created by dissolving PEO (M_w_~2 MDa, Sigma-Aldrich, Burlington, MA, USA) in DI water to a concentration of 0.05 wt% PEO. Fluorescent polystyrene (PS) particles (Thermo Scientific Inc., Waltham, MA, USA) with sizes of 2.1, 5.0, and 12 µm were introduced into the DI water and PEO solution, maintaining a concentration range of 0.05~0.1 wt% concentration. To prevent particle aggregation throughout the particle focusing and separation experiments, Tween 20 (Sigma-Aldrich, Burlington, MA, USA) was added into the suspensions at a concentration of 0.1 wt%, acting as a surfactant. A syringe pump (LEGATO 111, KD Scientific Inc., Holliston, MA, USA) was employed to inject the sample and sheath fluid into the microfluidic device. During the separation experiments, a total flow rate of 210 µL/min was maintained, leading to flow rate ratios (sample flow to sheath flow) of 1:2, 1:4, and 1:6.

The measurement locations were 10, 30, and 50 mm downstream from the inlet, and the pattern of particle focusing was captured using a CMOS camera (Suzhou ZWO Co., Ltd., Suzhou, China) on an optical microscope (BX-60, Olympus, Tokyo, Japan). Specifically, different bandpass filters were mounted on the camera lens, selected based on the characteristic emission wavelength of each type of fluorescent particle (i.e., blue for 2.1 µm, green for 5 µm, and red for 12 µm). Subsequently, images of the fluorescent particles were captured from both the top and side of the microchannel, utilizing a long exposure time ranging from 700 to 800 ms to analyze the particle focusing and separation phenomena. In this setup, only the streamlines for one type of fluorescent particle can be imaged per trial, despite the simultaneous presence of three differently sized particles at the time of capture. The subsequent processing and analysis of the acquired images were performed using the open-source ImageJ software 1.52a (NIH, New York, NY, USA), wherein the fluorescent intensity profiles across the microfluidic channel were obtained and fitted with a Gaussian distribution.

The separation performance was calculated by monitoring the particle movement at the outlet of the microchannel using a microscope system equipped with a high-speed camera (Phantom VEO-E 310L, Vision Research Inc., Wayne, NJ, USA), operating at 4000~6000 fps (frames per second). The number of each particle was counted using the Phantom Camera Control (PCC) application, where particles were manually traced from frame to frame. The separation efficiency, recovery ratio, and enrichment factor were defined as follows [24].
(1)$$\mathrm{Separation}\;\mathrm{efficiency}=\frac{N_{target}^{out1}}{N_{target}^{out1}+N_{target}^{out2}}$$
(2)$$\mathrm{Recovery}\;\mathrm{ratio}=\frac{N_{target}^{out1}}{N_{target}^{in}}$$
(3)$$\mathrm{Enrichment}\;\mathrm{factor}=(\frac{N_{target}^{out1}}{N_{non-target}^{out1}})/(\frac{N_{target}^{in}}{N_{non-target}^{in}})$$

### 2.3. Principle of Particle Separation via Co-Flow

While the particles in Newtonian fluids experience an inertial lift force (***F_L_***) and drag force (***F_D_***), the particles in viscoelastic fluids receive not only an inertial force (***F_L_***) but also an elastic force (***F_E_***) according to the diameter (*a*) of the particle [25,26]. Therefore, particle migration in viscoelastic fluid is dictated by both inertial and viscoelastic effects, and the particle trajectories and equilibrium positions under fluid flow are determined by the mutual interplay of these forces [26].

The inertial lift force (***F_L_***), responsible for the lateral migration of particles to equilibrium positions, is the combined effect of the shear gradient lift force and the wall lift force. The shear gradient lift force drives the particles near the channel center toward the wall, whereas the wall lift force pushes the particles close to the wall toward the channel center. The inertial lift force can be expressed as:(4)$$F_{L}=\frac{\rho_{f}{V_{m}}^{2}a^{4}}{{D_{h}}^{2}}f_{L}(Re,x_{c})$$
where $\rho_{f}$ is the fluid density, *V_m_* is the average velocity of the channel flow, $f_{L}(Re,x_{c})$ is the lift coefficient of the net inertial lift force, and *Re* is the Reynolds number.

The elastic force (***F_E_***) is mainly due to the distribution of the first normal stress (*N*_1_), which contributes to the lateral particle migration. The elastic force, originating from an imbalance in the distribution of *N*_1_ over the particle size [27], can be expressed as:(5)$$F_{E}=C_{eL}a^{3}\nabla N_{1}=-2C_{eL}a^{3}\eta_{p}\lambda \nabla {\dot{\gamma }}^{2}$$
where *C_eL_* is the non-dimensional elastic lift coefficient, $\eta_{p}$ is the polymeric contribution to the solution viscosity, λ is the relaxation time of the fluid, and $\dot{\gamma }$ is the local shear rate.

When a particle moves through a fluid or when the fluid flows past a particle, a viscous drag force (***F**_D_***) appears. This force arises from the velocity discrepancy between the particle and the fluid, and it can notably influence particle migration. This phenomenon can be expressed in a uniform Stokes flow as the following equation [28]:(6)$$F_{D}=3\pi \mu_{f}a(v_{f}-v_{p})$$
where $\mu_{f}$ is the dynamic viscosity of the fluid, and *v_f_* and *v_p_* represent the velocities of the fluid and particles, respectively.

The blockage ratio (*β* = *a*/*D_h_*), which relates the particle diameter (*a*) to the hydraulic diameter of the channel (*D_h_*), serves as a crucial parameter dictating the elasto-inertial particle focusing tendency in viscoelastic fluid [29]. Therefore, particle migration and focusing are more pronounced and concentrated at a high blockage ratio (*β* > 0.07) since the force exerted on the particle due to normal stress is directly proportional to the cube of the particle diameter [29,30].

Figure 2 shows the schematic view of size-based particle separation using a co-flow system with a T-shaped microchannel. For Case 1, as shown in Figure 2a, the migration direction in a co-flow system of Newtonian and viscoelastic fluids was from viscoelastic fluid to Newtonian fluid [12], which was dependent on the viscoelasticity of the viscoelastic fluid. However, for Case 2, there were only inertial forces exerted on the particles, as shown in Figure 2b.

For Case 1, the elastic force (***F_E_***) could transport particles across the interface between the viscoelastic fluid and Newtonian fluid and into the Newtonian fluid regime if the elastic force (***F_E_***) was sufficiently strong enough to counteract the opposing inertial forces, such as ***F_L_*** and ***F_D_***. Furthermore, the lateral migration speed of particles needs to be sufficient for particles to move into the Newtonian flow, which correlates to the square of their diameter in the viscoelastic fluid [14]. After particles move out of the viscoelastic fluid, the elastic force (***F_E_***) disappears. However, particles can still move further toward an equilibrium position even under the Newtonian fluid flow due to the inertia of the particles. At this time, larger particles with a high blockage ratio (*β* > 0.07) move much faster than smaller particles because the migration speed in a Newtonian fluid is proportional to the cube of the particles’ diameters [31]. However, for particles with a low blockage ratio (*β* < 0.07), the elastic force is not strong enough to move particles to the Newtonian fluid, so they remain within the viscoelastic fluid.

## 3. Results and Discussion

### 3.1. Particle Focusing Position in Newtonian and Viscoelastic Fluids

First, we investigated the particle-focusing position in both Newtonian and viscoelastic fluids. The particle-focusing positions in the microchannel with a T-shaped cross-section were identified without the sheath fluid flow for the various flow rates along the channel length. Figure 3 shows the fluorescent images and intensity graphs of particle-focusing under viscoelastic fluid flow, as presented in the previous study [23]. At the same time, the particle focusing positions under Newtonian fluid flow could be evaluated (see Figure 4 and Appendix A) prior to the particle separation experiments using the sheath flow (comprising viscoelastic and Newtonian fluids).

Particle images captured separately from the top and side of the microchannels allowed for the assessment of the particle positions in the cross-plane of the channel. Particles were focused at the bottom center of the T-shaped microchannel under viscoelastic fluid flow. However, focusing positions varied depending on the particle size; 12 μm particles exhibited three distinct focusing positions, whereas 5.0 and 2.1 μm particles displayed focusing bands near the centerline. When the particle focusing experiments with a square microchannel having the same hydraulic diameter (*D_h_* = 72 μm) as the microchannel with a T-shaped cross-section were conducted, 2.1 μm particles did not focus to the centerline due to the low blockage ratio (*β* = 0.03) (Appendix A). It proved that two reflex angles (270°) of the T-shaped cross-section helped to focus particles with a low blockage ratio [22,23].

In contrast, under Newtonian fluid flow, particles were two-dimensionally focused in the T-shaped cross-section of the microchannel, as shown in Figure 4. To be specific, 12 μm particles presented three focusing positions around the center of the bottom region, and 5.0 μm particles showed four focusing points near the corner of the channel cross-section with the reflex angle. However, 2.1 μm particles did not exhibit pronounced focus. Instead, a slight focusing was observed in a position similar to that of the 5.0 μm particles.

### 3.2. Effects of the Sample-to-Sheath Flow Rate Ratio

To validate the effects of flow conditions on separation performance, a mixture of PS fluorescent particles with diameters of 2.1 μm (blue), 5.0 μm (green), and 12 μm (red) was employed. The particle diameters were selected by considering the size of the target cells. In this study, blood cells were selected as target cells, which consisted of white blood cells, red blood cells, and platelets whose sizes approximately corresponded to 12, 5.0, and 2.1 μm. These particles had concentrations of 9.8 × 10^9^, 7.2 × 10^8^, and 5.2 × 10^7^ particles/mL, respectively. They were tested under two sample/sheath flow conditions: viscoelastic/Newtonian fluids (Case 1) and Newtonian/Newtonian fluids (Case 2). The sample fluid, which contained a mixture of particles of three different sizes, was injected from Inlet 1, while the sheath fluid was introduced from Inlet 2.

For Case 1, a sample flow of PEO solution containing particles and a sheath flow of DI water were employed. The sample-to-sheath flow rate ratio was varied, with the total flow rate ranging from 30 to 250 µL/min. Figure 5 presents the fluorescent images of particle movement at the channel outlet for the flow rate ratios of 2, 4, and 6, respectively. As the flow rate ratio increased (with the total flow rate fixed at 210 µL/min), the interface between the PEO solution (sample flow) and the DI water (sheath flow) shifted closer to the sidewall. That is, particles of different sizes were initially aligned near the bottom side walls of the T-shaped channel via sheath flow.

As the fluid flowed in the described manner, 12 μm particles with a high blockage ratio (*β* = 0.17) migrated from within the boundary of the PEO solution (viscoelastic fluid) and flowed along the DI water (Newtonian fluid) near the outlet of channels. However, 2.1 μm particles with a low blockage ratio (*β* = 0.03) remained in the viscoelastic fluid regime due to the lack of elastic force preventing their migration to the Newtonian fluid area. Meanwhile, 5.0 μm particles, possessing a blockage ratio greater than 0.07, were able to cross the PEO solution boundary. However, the force acting on them, which was proportional to the particle size, resulted in a different migration speed at the center of the channel as compared with the 12 μm particles. Consequently, there was a difference in position between the two particles. Specifically, 12 μm particles reached the centerline within a 60 mm T-shape microchannel and exited through Outlet 3, whereas 5.0 μm particles, not reaching the centerline, exited through Outlets 2 and 4. The 2.1 μm particles, which remained near the sidewalls of the microchannel, exited through Outlets 1 and 5. Based on these migrating behaviors of particles, it may be feasible to separate particles of three distinct sizes at the outlet of the channel. When the flow rate ratio exceeded 2, 2.1 µm particles were efficiently separated (>90%) from the larger particles. A mixture of 2.1, 5.0, and 12 μm particles initially injected into the sidewall of the channel were successfully separated at the microchannel outlet. When the flow rate was 6 or more, all particles could be separated from each other at the same time.

For Case 2, DI water served as the sample fluid, ensuring comparison of the particle separation efficiency under identical experimental conditions (total flow rate of 210 µL/h and flow rate ratios from 2 to 6). Both the sample flow containing particles and the sheath flow were comprised of DI water in this case. Figure 6 presents the fluorescent images of particle movement at the channel outlet for the flow rate ratios of 2, 4, and 6. Similar to Case 1, as the flow rate ratio increased, the interface between the DI water sample flow and DI water sheath flow moved closer to the sidewall. Eventually, all particles experienced a force directing them away from the wall. However, in contrast to Case 1, only the largest 12 μm particles shifted toward the channel center, and they did not entirely reach the centerline. Moreover, 5.0 μm particles, similar to 2.1 μm particles, did not experience enough force to move them to the centerline, leading them to remain close to the sidewall.

In both cases, the interface position between the sample and the sheath fluids was the same. As a result, the wall lift force, which pushed particles away from the wall, acted in the same direction. However, Case 1 induced an additional elastic force, which was absent in Case 2. Such an additional force drove the particles toward the center of the channel. Therefore, while it was possible to separate the three different-sized particles under viscoelastic/Newtonian fluid flow, it was challenging to separate them under Newtonian/Newtonian fluid flow with identical flow conditions.

For Case 1, the elastic force acted in the same direction as the wall lift force, both pushing the particles away from the wall. When the particles crossed the interface between the two fluids, the residual elastic force from the viscoelastic fluid further pushed the particles toward the channel center.

### 3.3. Separation Efficiencies, Recovery Ratio, and Enrichment Factor

Figure 7a,b show the separation of a mixture of 2.1 μm (blue), 5.0 μm (green), and 12 μm (red) particles under two sample/sheath flow conditions: viscoelastic/Newtonian (Case 1) and Newtonian/Newtonian (Case 2). Under the viscoelastic/Newtonian fluid flow condition (Case 1), 2.1 μm particles remained on the sidewall of the channel and exited through Outlets 1 and 5, since they were initially aligned at the beginning of the channel. On the other hand, 5.0 μm particles, influenced by the appropriate elastic force (***F_E_***) and wall lift force (***F_L_***) from the viscoelastic fluid, moved toward Outlets 2 and 4. The 12 μm particles rapidly migrated to the centerline of the channel across the laminar stream from the viscoelastic to Newtonian fluid and thus exited through Outlet 3.

In the Newtonian/Newtonian fluid flow condition (Case 2), 2.1 μm particles behaved similarly as in Case 1 by remaining on the channel sidewall. The 5.0 μm particles, without the influence of the elastic force from the viscoelastic fluid, hardly migrated toward the channel center. As a result, they exited through Outlets 1 and 5, as the 2.1 μm particles did. As shown in Figure 7c, with the flow rate ratio of 1:6 for Case 1, the separation efficiency for each of the three distinct particles exceeded 95%. On the other hand, when DI water was used as the sample solution instead of PEO solution, the separation efficiencies of 5.0 μm and 12 μm particles significantly decreased from 94% and 95% to 10% and 13%, respectively (flow rate ratio = 1:6). The recovery ratio and enrichment factor are presented for both Case 1 and Case 2 at the flow rate ratio of 6 in Appendix A. For the separation of three differently sized particles, the 12 μm particles exhibited the highest recovery ratio (>93%) and enrichment factor (>100) as compared with the other particles.

Figure 8 presents the separation efficiencies for three combinations of particles (2.1 & 5.0 μm, 2.1 & 12 μm, and 5.0 &12 μm) for both Case 1 and Case 2 at the flow rate ratio of 6. More detailed data are presented in Table 2 and Appendix A. When the particles were introduced into the PEO solution (Case 1) and the flow rate ratio varied from 2 to 6, both the separation efficiency and recovery ratio for all particle combinations increased. The separation efficiency and recovery ratio improved by up to 95% and 93%, respectively. Furthermore, for the mixtures of 2.1 and 5.0 and 2.1 and 12 μm, the 2.1 μm particles exhibited the highest separation efficiency (>97%) and recovery ratio (>95%) as compared with the other particles. In the separation of 2.1 and 12 μm particles, the 12 μm particles exhibited a very high separation efficiency (95%) and enrichment factor (>700) in Case 1 (particles in PEO solution), which were higher than those in Case 2 (particles in DI water, >12% and >23). This suggests that the 2.1 μm particles rarely reached Outlet 3. Moreover, a larger difference in particle size corresponded to better separation efficiency, recovery rate, and enrichment factor.

On the other hand, the 2.1 μm particles exhibited a slightly higher separation efficiency in Case 2 than in Case 1. While the particles in Case 1 were affected by both elastic and inertial forces, those in Case 2 were influenced solely by the inertial force directed toward the channel center. Therefore, the 2.1 μm particles in Case 2 remained closer to the wall side of the channel, facilitating easier separation.

It is known that an increased number of particles induces a particle-particle interaction that interferes with particle migration [32]. This phenomenon may explain why the separation efficiency for the three particles was slightly lower than that for the two particles in Case 1. Nevertheless, the fabricated microfluidic device with a T-shaped microchannel considered in this study could be suitable for separating either two or three differently sized particles with a better efficiency and recovery ratio using the co-flow of viscoelastic/Newtonian fluids.

## 4. Conclusions

In this study, we demonstrated a continuous and size-dependent particle separation using a co-flow of viscoelastic and Newtonian fluids in a straight channel with a T-shaped cross-section, whose fabrication processes were proposed in a previous study. When using the co-flow of the viscoelastic sample fluid and Newtonian sheath fluid (Case 1), the highly efficient and continuous separation of three particles of different sizes could be achieved. Due to the co-flow of viscoelastic and Newtonian fluids, large particles moved from the viscoelastic fluid toward the center of the channel, but small particles stayed at the interface between two fluids near the side of the channel. Moreover, two reflex angles (270°) of the T-shaped cross-section helped the separation of three differently sized particles even at high flow rates when compared with the square microchannel. As the flow rate ratio increased, the separation efficiency and recovery ratio increased significantly, achieving a high separation efficiency (>98%), recovery ratio (>97%), and enrichment factor (>700). With such a simple structure and excellent particle separation performance, the fabricated microfluidic devices with a T-shaped microchannel have great potential in particle separation processes for lab-on-a-chip and biomedical applications, such as cell separation. These particle sizes correspond to the size of blood cells (white blood cells, red blood cells, and platelets), so this separation technique could be utilized for blood cell separation.

## Figures and Tables

**Figure 1 micromachines-14-01863-f001:**
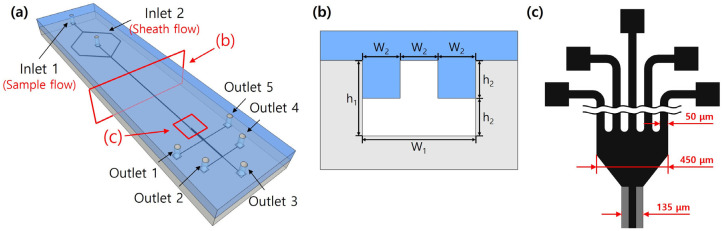
Schematic illustration of the microchannel with a T-shaped cross-section. (**a**) 3D view, (**b**) cross-sectional view, (**c**) geometry of 5-way outlets.

**Figure 2 micromachines-14-01863-f002:**
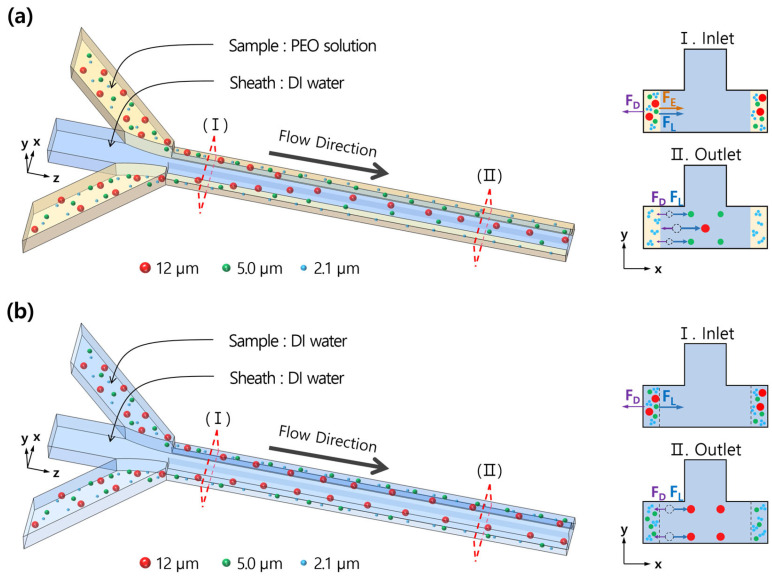
Schematic illustration of size-based particle separation using a co-flow system with a T-shaped microchannel. (**a**) Case 1 (Newtonian/Viscoelastic fluids), (**b**) Case 2 (Newtonian/Newtonian fluids).

**Figure 3 micromachines-14-01863-f003:**
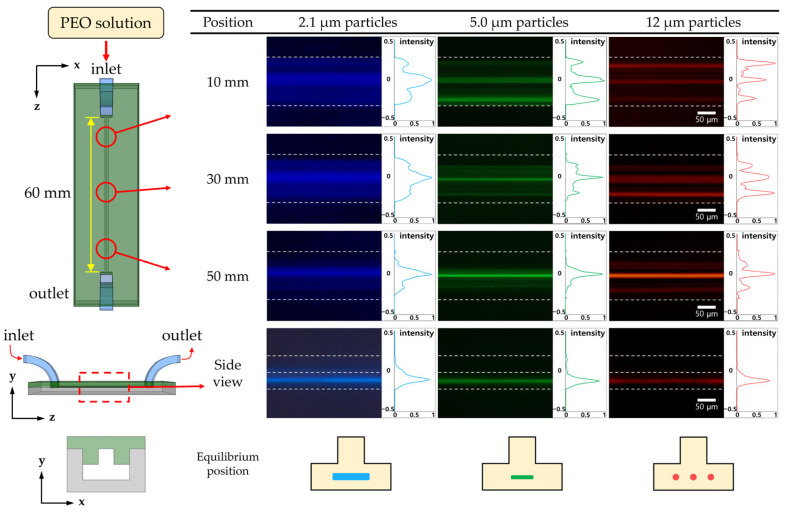
Fluorescent images and intensity graphs of particle focusing along the channel length under viscoelastic fluid flow. To find the equilibrium focusing position, fluorescent images from top and side views were captured, respectively (flow rate: 100 μL/min).

**Figure 4 micromachines-14-01863-f004:**
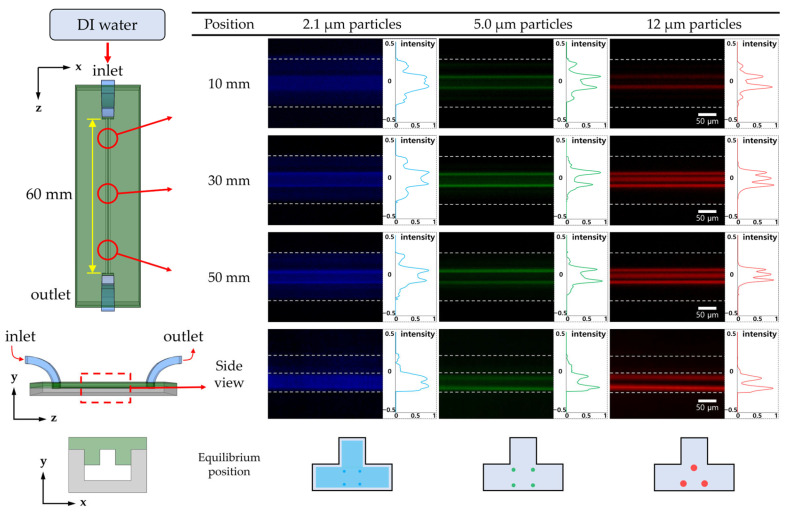
Fluorescent images and intensity graphs of particle focusing along the channel length under Newtonian fluid flow. To find the equilibrium focusing position, fluorescent images from top and side views were captured, respectively (flow rate: 100 μL/min).

**Figure 5 micromachines-14-01863-f005:**
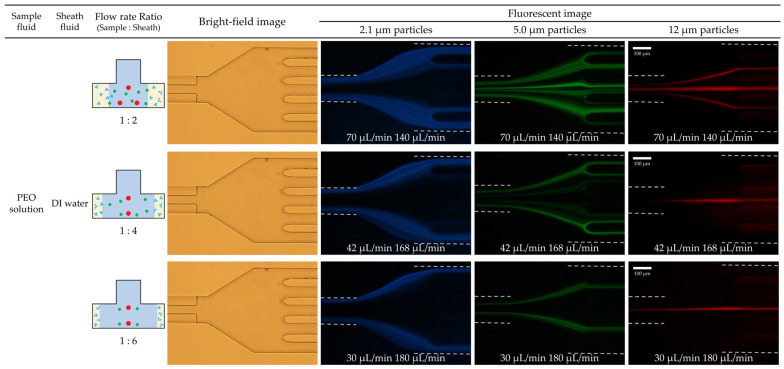
Particle movement according to the change in flow rate ratio from 2 to 6 with viscoelastic/Newtonian flow and a total flow rate of 210 µL/min. Experimental results of particle separation in a microchannel with a T-shaped cross-section at sample flow (30 µL/min) and sheath flow (180 µL/min) were presented. Bright-field image showed the shape of 5-way outlets, and fluorescent images show the different exits of 2.1, 5.0, and 12 µm particles at 5-way outlets.

**Figure 6 micromachines-14-01863-f006:**
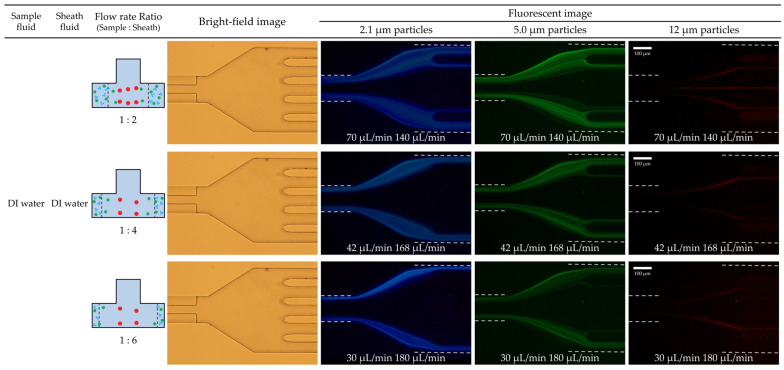
Particle movement according to the change in flow rate ratio from 2 to 6 with Newtonian/Newtonian flow and a total flow rate of 210 µL/min. Experimental results of particle separation in a microchannel with a T-shaped cross-section at sample flow (30 µL/min) and sheath flow (180 µL/min) were presented. Bright-field image showed the shape of 5-way outlets, and fluorescent images show the different exits of 2.1, 5.0, and 12 µm particles at 5-way outlets.

**Figure 7 micromachines-14-01863-f007:**
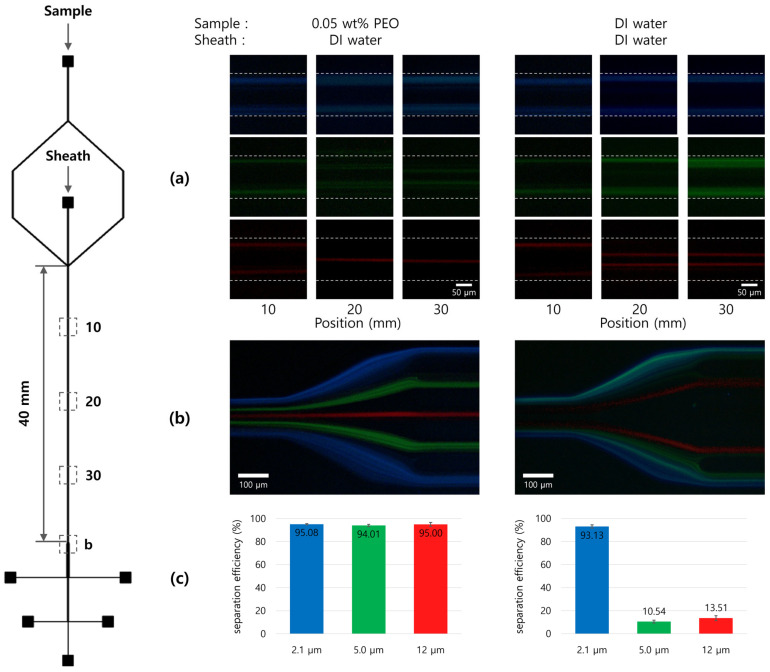
Separation of a mixture of 2.1 μm (blue), 5.0 μm (green), and 12 μm (red) particles under two sample/sheath flow conditions: viscoelastic/Newtonian and Newtonian/Newtonian. (**a**) Fluorescent images of each particles along 10, 20, 30 mm in the main microchannel, (**b**) fluorescent images of three particles at the 5-way outlets, (**c**) graph of separation efficiency for three distinct particle mixture.

**Figure 8 micromachines-14-01863-f008:**
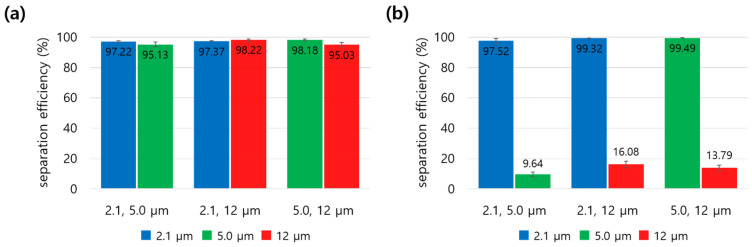
Separation efficiencies of 2.1 μm, 5.0 μm, and 12 μm particles under two sample/sheath flow conditions: (**a**) viscoelastic/Newtonian, (**b**) Newtonian/Newtonian. The total flow rate and flow rate ratio were 210 µL/min and 6.

**Table 1 micromachines-14-01863-t001:** Experimental conditions for injecting the sample into inlet 1 in the microchannel with a T-shaped cross-section.

Case	Inlet 1	Inlet 2
1	PEO solution + Particle	DI water
2	DI water + Particle	DI water
3	PEO solution + Particle	PEO solution
4	DI water + Particle	PEO solution

**Table 2 micromachines-14-01863-t002:** Separation efficiencies, recovery ratio, and enrichment factor of 2.1 μm, 5.0 μm, and 12 μm particles at various flow rate ratios of sample and sheath ranging from 2 to 6, with a fixed total flow rate of 210 µL/min.

Case	Factor	1:2						1:4						1:6					
		(2.1, 5.0 μm)		(2.1, 12 μm)		(5.0, 12 μm)		(2.1, 5.0 μm)		(2.1, 12 μm)		(5.0, 12 μm)		(2.1, 5.0 μm)		(2.1, 12 μm)		(5.0, 12 μm)	
		2.1 μm	5.0 μm	2.1 μm	12 μm	5.0 μm	12 μm	2.1 μm	5.0 μm	2.1 μm	12 μm	5.0 μm	12 μm	2.1 μm	5.0 μm	2.1 μm	12 μm	5.0 μm	12 μm
Case 1	separation efficiency(%)	56.88	81.87	97.17	38.46	60.62	33.33	77.68	75.23	99.45	75.61	92.04	84.03	97.22	95.13	97.37	98.22	98.18	95.03
	recovery ratio(%)	55.21	78.95	94.22	36.36	58.69	31.86	75.42	72.42	96.36	72.09	89.24	80.65	94.88	93.06	95.26	97.65	95.71	94.51
	enrichment factor	3.16	1.89	1.62	13.24	0.92	0.84	3.16	3.34	4.14	136.4	5.82	10.45	19.90	34.33	53.98	719.2	19.35	53.15
Case 2	separation efficiency(%)	67.84	38.17	95.88	44.35	94.30	39.02	87.38	24.24	97.60	26.71	97.38	21.23	97.52	9.64	99.32	16.08	99.49	13.79
	recovery ratio(%)	66.08	37.39	93.50	43.22	92.50	38.10	83.22	22.95	94.17	25.49	94.01	20.13	95.09	9.39	97.10	15.75	96.53	13.51
	enrichment factor	1.09	1.19	1.72	10.75	1.55	6.81	1.16	1.91	1.35	10.99	1.26	7.97	1.08	3.87	1.18	23.69	1.14	27.23

## Data Availability

Not applicable.

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
