# Peer review of "Particle Separation in a Microchannel with a T-Shaped Cross-Section Using Co-Flow of Newtonian and Viscoelastic Fluids"

_micromachines, 2023, doi:10.3390/mi14101863_

Round 1

Reviewer 1 Report

In this paper, the particle separation phenomenon in the micro-channel is analyzed on the basis of the T-shaped cross-section microchannel designed in the previous study. The co-flow experiments of Newtonian/viscoelastic fluid and Newtonian/Newtonian fluid are conducted and compared. The Newton/viscoelastic fluid successfully separated 3 particles of different sizes. Large particles (12 μm) moved to the center of the channel, while small particles (2.1 μm) stayed at the side wall interface of the channel. It is proved that the co-flow of Newton/viscoelastic fluid in T-section microchannel can effectively separate particles of different sizes at the same time, and the separation efficiency and recovery rate are significantly improved with the increase of flow rate ratio. However, there are some deficiencies should be revised:

1.      In the abstract, experimental results are not compared, and the results relative to which experiments are not clear enough.

2.      The reference to CTCs in the second paragraph of the introduction should be accompanied by a paper citation.

3.      The experiments in the paper all compare the viscous/Newtonian fluid and Newtonian/Newtonian fluid, but they are not reflected in the title of the paper.

4.      On Page 4 , the reasons for selecting the flow conditions and particle diameter are incomplete, which have an important influence to the experiment results.

5.      The citation [13] is not consistent with the full text.

6.      The description of microchannel dimensions in Figure 1 is incomplete.

7.      (a), (b), (c) and (d) in Figure 6 are not marked.

8.      (3) is not explained in Figure 8.

9.      (a) (b) is not illustrated in Figure 9.

10.   On Page 11, line 366 , the follow-up content of "It is due to the fact that" is not completed.

The English should be revised carefully.

Reviewer 2 Report

 This work is about the particle separation in microchannel with T-shaped cross-section using co-flow of Newtonian and viscoelastic fluids. The research topic is interesting, and the manuscript provides substantial results and discussions. However, the description of some experimental results in the article is insufficient. For example, there is a lack of experiments and data comparing microchannels with T-shaped or square cross-sections, and the lack of observation images of some critical positions in microchannels. There are some suggestions below:

1. In the "Introduction" section, lines 47-52 need to supplement relevant references.

2. In the " Materials and Methods” section, lines 98-99 the specific sizes of h2, w2, and W3 were not provided, and it is recommended to supplement them.

3. In the "Results and Discussion" section, as mentioned in the "Introduction" section of the article, "Under the action of viscoelastic fluids, particles in the T-shaped cross-section microfluidic channel focus well near the obtuse angle of its cross-section." It is recommended to supplement the analysis of experimental results in this section based on the specific size of the structure to determine whether particles focus near the obtuse angle of the T-shaped cross-section microfluidic channel.

4. In the "Results and Discussion" section, for all the images in the manuscript, only Figure 8 (b) shows images that can simultaneously see three types of fluorescent particles, while other images can only see a single type of fluorescent particles. However, Figure 8 (b) does not demonstrate the process of mixing and separating these three types of fluorescent particles, so the images in the manuscript cannot effectively support the separation effect. Therefore, it is recommended to modify the image in Figure 8 by adding three observed positions: the first focal area where the sample and shear phase flow in, and the two outlet bifurcation areas where the three particles flow out after separation. Based on the collection images of the three positions, add the initial focusing images of the three uniformly mixed solutions and particle dispersion phases in Figure 8, and the three particle separation images flowing to different outlet bifurcation regions.

5. In the "Conclusion" section mentioned that "T-shaped cross-section helped the separation of three different sized particles even at high flow rates when compared with the square microchannel". However, I did not see experiments and data comparing microchannels with T-shaped or square cross-sections in the “results and discussion” section. Therefore, it is recommended to supplement relevant experimental data and results.

Reviewer 3 Report

The authors report experimental work on particle separation in previously developed microchannels. They presented various experimental results to verify the idea of ​​the proposed microchannel system. They performed particle separation experiments using Newtonian/viscoelastic fluids and coflows of Newtonian/Newtonian fluids. They showed particle separation of three different sizes for different flow rate ratios between Newtonian and viscoelastic fluids. They also demonstrated the high separation efficiency and recovery rate of the system. Overall, this work appears premature and requires careful revision. In particular, figure captions must be strengthened to provide a sufficient explanation of the figure. I therefore recommend that this work be carefully rewritten and revised. Here are some more detailed comments:

1. Figure captions

The overall figure caption is too short, and in some cases, figure captions are missing. For example, Figure 8 does not have a caption for Figure 8c. Authors should carefully read and edit figure captions.

2. page 6, line 222

Figures 3, 4, and 5 are top view only. A side view of the microchannel is required. Authors should note that they should refer to the supplementary material for side view information. Why don’t the authors include a side view in the main figure?

3. Figures 3-5 can be combined into one figure for a more concise presentation.

4. Confusing Figures (Figs. 6 and 7)

It's very confusing as there is no explanation of the outlet geometry. Include 5-way outlet geometry information (channel geometry, etc.).

5. What is the 5-way outlet? A detailed explanation is needed

I recommend that this work be carefully rewritten and revised especially for the figure captions.

Round 2

Reviewer 2 Report

The author has responded to all the suggestions and made corresponding modifications. I suggest to accept the manuscript in present form.

Reviewer 3 Report

All the issues raised by the reviewer have been properly addressed.